# Myc-Interacting Zinc Finger Protein 1 (Miz-1) Is Essential to Maintain Homeostasis and Immunocompetence of the B Cell Lineage

**DOI:** 10.3390/biology11040504

**Published:** 2022-03-24

**Authors:** Eva-Maria Piskor, Julie Ross, Tarik Möröy, Christian Kosan

**Affiliations:** 1Institute of Biochemistry and Biophysics, Center for Molecular Biomedicine (CMB), Friedrich Schiller University, Hans-Knöll-Str. 2, 07745 Jena, Germany; eva-maria.piskor@uni-jena.de; 2Hematopoiesis and Cancer Unit, Institut de Recherches Cliniques de Montréal (IRCM), 110 av. Des Pins O, Montréal, QC H2W 1R7, Canada; julie.ross@ircm.qc.ca (J.R.); tarik.moroy@ircm.qc.ca (T.M.); 3Département de Microbiologie, Infectiologie et Immunologie, Université de Montréal, 2900, boul. Édouard-Montpetit, Montréal, QC H3T 1J4, Canada; 4Division of Experimental Medicine, McGill University, 801 Sherbrooke St. W., Montréal, QC H3A 0B8, Canada

**Keywords:** Miz-1, aging, B cell development, B cell maturation

## Abstract

**Simple Summary:**

The immune system of mice and humans acts against pathogenic threats and intrinsic risks such as cancer. B cells, as antibody-producing cells, provide the ability to specifically target these risks. However, aging leads to a progressive loss of this ability and molecular causes of the gradual loss of immunocompetence remain unknown. Using genetically modified mice, we unravel the transcription factor Miz-1 as a key player of B cell aging during bone marrow lymphopoiesis and peripheral maturation. This enables the investigation of B cell-specific aging mechanisms and how to counteract them for therapeutic approaches to improve immunocompetence in the elderly.

**Abstract:**

Aging of the immune system is described as a progressive loss of the ability to respond to immunologic stimuli and is commonly referred to as immunosenescence. B cell immunosenescence is characterized by a decreased differentiation rate in the bone marrow and accumulation of antigen-experienced and age-associated B cells in secondary lymphoid organs (SLOs). A specific deletion of the POZ-domain of the transcription factor Miz-1 in pro-B cells, which is known to be involved in bone marrow hematopoiesis, leads to premature aging of the B cell lineage. In mice, this causes a severe reduction in bone marrow-derived B cells with a drastic decrease from the pre-B cell stage on. Further, mature, naïve cells in SLOs are reduced at an early age, while post-activation-associated subpopulations increase prematurely. We propose that Miz-1 interferes at several key regulatory checkpoints, critical during B cell aging, and counteracts a premature loss of immunocompetence. This enables the use of our mouse model to gain further insights into mechanisms of B cell aging and it can significantly contribute to understand molecular causes of impaired adaptive immune responses to counteract loss of immunocompetence and restore a functional immune response in the elderly.

## 1. Introduction

The general cause of aging is widely considered to be a time-dependent accumulation of cellular damage, resulting in the progressive loss of physiological integrity [1,2,3]. Here, besides controlling infections, the immune system is involved in controlling malignancies, tissue homeostasis and repair [4,5,6]. It undergoes progressive transformations and the age-related decrease in immunological competence is often referred to as immunosenescence [7]. The progressive deterioration leads to diminished efficiency, reduced immunological sensitivity and a low-grade inflammatory environment predisposing to higher susceptibility to autoimmune disease or cancer development [5,7,8,9,10]. B cells are an essential part of preserving integrity of the body, as their activation and differentiation into antibody-secreting cells (ASCs) provide the ability to specifically target an encountered threat such as infection or cancer. B cell aging is characterized by accumulation of antigen-experienced and undefined cell phenotypes in aged mice, while the naïve compartment, as well as repertoire and diversity, is reduced [11,12,13]. The privation of optimal B cell diversity is accompanied by reduced affinity of antibody responses [14,15].

During aging, B cells become less responsive to cytokine signaling. Common lymphoid progenitors (CLPs) and pro-B cells grow less in response to IL-7 stimulation in vitro, although IL-7R surface expression is not altered [16,17,18]. Furthermore, function and proliferation of old BM mesenchymal stromal cells (MSC) are impaired [19]. An age-related decrease in the function of bone marrow stromal cells in correlation with reduced B lymphopoiesis is related to impaired release of IL-7 [20].

Additionally, B cell-intrinsic mechanisms contribute to a reduction in antibody affinity. Upregulation of Id2, an E2A inhibitor, increases with age and correlates with reduced IGHV and D alterations in aged transitional and naïve B cells and supports diminished antibody responses of old B cells [21,22,23]. Compromised E2A expression and impaired CSR predispose to compromised antibody responses in old B cells [21,24,25,26,27,28]. At the same time, a progressive accumulation of age-associated B cells (ABCs) was described. ABCs show accelerated development in autoimmune conditions such as systemic lupus erythematosus or rheumatoid arthritis [29,30]. They are potent TNF-α producers predisposing a pro-inflammatory state, which was shown to reduce B cell fitness, activation and response [12,31,32]. Accumulation of terminally differentiated cells and serum immunoglobulins determines a homeostatic pressure on the bone marrow B lymphopoiesis rate [33]. Studies showed decreasing rates of B lymphopoiesis in old mice and slower output kinetics to SLOs [15,34,35]. Altered multifactorial transcriptional regulation events further amplify the attenuation. In aged mice, Id2 expression increases and the magnitude of E47 reduction, the product of E2A, in pro-B cells correlates with the severity of pre-B cell loss [22,36,37]. E2A is a key regulator of *EBF1*, *SLC* and IGK locus accessibility [38,39,40]. Its reduction and subsequent combinatorial decrease in RAG and SLC expression alter pre-BCR and BCR surface expression due to distorted accessibility of the Ig chain loci and result in reduced B cell development in aged mice [41,42]. This correlates with reduced *Rag1/2* expression and decreased VDJ recombination [43,44,45].

The transcription factor Myc-interacting zinc finger protein 1 (Miz-1) is a key regulator of early B cell development, germinal center (GC) reaction and differentiation of memory B cells [46,47]. Miz-1 was originally described as a binding partner of the proto-oncoprotein c-Myc and transcriptional regulation by Miz-1 is dependent on homo- or heterodimerization with interaction partners via its POZ-domain [46,48,49,50,51]. However, its expression is ubiquitous in all mouse tissues. In B cells, Miz-1 regulates IL7 signaling to ensure early B cell survival and development, independent of its co-factor Myc [46]. However, during GC reactions, a functional Miz-1/Myc complex is important for cell proliferation and memory B cell formation [47].

Here, we show that Miz-1-deficient mice show signs of premature aging in the B cell compartment. Miz-1 is essential to control B cell development in aging mice, for counteracting premature reduction in naïve B cells and accumulation of antigen-experienced B cells in peripheral lymphoid organs.

## 2. Materials and Methods

### 2.1. Mice

C57BL/6JRj wildtype and genetically modified mice with C57BL/6JRj background were bred and housed under specific pathogen-free conditions in individual ventilated cages in the Experimental Biomedicine Unit of the University of Jena in Germany according to European guidelines (2010/63/EU). Mice had no restrictions on food and water. Mice were bred according to registration number 02-053/16 approved by the TLV. Mb1^Cre^ (CD79a^tm1(cre)reth^; The Jackson Laboratory (with the kind permission of Prof. Dr. M. Reth)); Miz-1^fl^ (Zbtb17^tm1Cksn^; The Jackson Laboratory; Kosan et al., 2010); C57BL/6 (C57BL/6JRj; Janvier Lab) on C57BL/6 background were used in this study. Zbtb17^tm1Cksn^ mice have been described previously (Kosan et al. 2010). Animals with the Miz-1^fl/fl^ Mb1-Cre^tg^ genotype (hereafter Miz-1^ΔPOZ^) were used for analysis. Age-matched Miz-1^fl/fl^ mice which did not express Cre-recombinase (Miz-1^fl/fl^Mb1-Cre^wt^) were used as control animals. Mice were analyzed at 3, 12 and 24 months of age.

### 2.2. Flow Cytometry and Antibodies

Single-cell suspension of spleen, Peyer’s patches of the intestine and 1 femur were processed for flow cytometric analysis at time of autopsy. A maximum of 1 × 10^6^ cells were used for antibody staining. Staining was performed at 4 °C for 15 min in MACS buffer (PBS with 0.5% FCS and 2 mM EDTA). CD45R/B220 (RA3-6B2), CD19 (eBio1D3), CD21/CD35 (8D9), IgD (12-26c), IgM (eB121-15F9), CD23 (B2B3), IgA (mA-6E1), CXCR4 (2B11), CD4 (RM4-5), CD8a (53-6.7), Gr-1 (RB6-8C5), Ly6C (HK1.4) were from eBioscience/Invitrogen (Waltham, MA, USA). CD93 (AA4.1), T- and B-Cell Activation Antigen (GL7), CD43 (S7), IgD (11-26c.2a), CD95 (Jo-2), IgG1 (X-56) were from BD Biosciences (Franklin Lakes, NJ, USA). CD38 (90), IgD (11-26c.2a), CD19 (1D3), CD86 (GL-1), CD3 (500A2), CD11c (N418) were from Biolegend (San Diego, CA, USA). CD5 (REA-421), Siglec-F (REA798), F4/80 (REA126) were from Miltenyi Biotec B.V. &Co. KG (Bergisch Gladbach, Germany). FSC and SSC signals were acquired to eliminate dead cells and doublets. The resulting population was defined as living cells. If not otherwise indicated, percentages in plots are given as percentage per parent. Data were acquired with LSRFortessa^TM^ and BD FACSDIVA V8.0.1 (BD Bioscience, Franklin Lakes, NJ, USA). Cell sorting was performed with BD FACSAria™ Fusion. Analysis was performed using FlowLogic 700.2A (Inivai Technologies Pty. Ltd., Menton, Australia).

### 2.3. B Cell Isolation

Splenic B cells were purified with a Pan-B Cell Purification kit II (Miltenyi Biotec B.V. &Co. KG, Bergisch Gladbach, Germany) and B Cell Purification Kit (Miltenyi Biotec B.V. &Co. KG, Bergisch Gladbach, Germany) according to the manufacturer’s instructions.

### 2.4. RNA Isolation and Quantitative Real-Time PCR

For high-quality RNA, cells were lysed in TRIzol^TM^ Reagent (Thermo Fisher, Waltham, MA, USA) and RNA was isolated by phenol/chloroform extraction according to the manufacturer’s instructions. For quantitative real-time experiments, cDNA was generated from 1 µg RNA using the First Strand cDNA Synthesis Kit (Thermo Fisher, Waltham, MA, USA) according to the manufacturer’s instructions, and an equally mixed combination of oligo(dT)18 and random hexamer primers. For single reactions, PowerUp SYBR^®^ Green Mastermix (Applied Biosystems, Waltham, MA, USA) was combined with specific primers (200 pmol) and 5 ng cDNA. Reactions and negative controls were performed in technical triplets. Data analysis was carried out with the comparative ΔΔCt method.

### 2.5. Isolation of Intestinal Antibodies

Fecal samples were collected, weighed, snap-frozen with liquid nitrogen and stored at −80 °C. For extraction of antibodies, fecal pellets were resuspended in PBS supplemented with 0.5% Tween 20 and proteinase inhibitors (1×) at a ratio of 20% *w/v* and incubated for 30 min at 4 °C while rotating. Suspensions were centrifuged at 4 °C for 10 min at 10,000× *g* and supernatant was collected.

### 2.6. Enzyme-Linked Immunosorbent Assay

Antibody concentration from fecal samples was determined by ELISA with an IgA Mouse Uncoated ELISA Kit (Thermo Fisher, Waltham, MA, USA) according to the manufacturer’s instructions. A starting dilution of 100-fold was used for fecal samples.

### 2.7. Statistical Analysis

Statistical analysis and linear regression calculation were performed with Prism 9 (GraphPad Software, San Diego, CA, USA). Unless indicated otherwise, data are expressed as mean + SD. Unless indicated otherwise, two-tailed unpaired Student’s *t*-test or ANOVA was used to calculate *p* values. Significant difference was defined as *p* ≤ 0.05 (*), *p* ≤ 0.01 (**), *p* ≤ 0.001 (***) and *p* ≤ 0.0001 (****).

## 3. Results

### 3.1. Deletion of Miz-1 Reduces the B Cell Differentiation Potential in the Bone Marrow

To address the question of whether deletion of Miz-1 effects aging of B lymphocytes, we analyzed the B cell compartment of Miz-1^ΔPOZ^ and control mice at young (3 months) and aged (12 and 24 months) stages. Data from flow cytometry demonstrate that, in comparison to control animals, Miz-1^ΔPOZ^ mice have less than 50% of B cells (B220^+^CD19^+^) in the bone marrow at all analyzed timepoints (Figure 1A,B). B cell frequency declines with age in both groups, and control animals also show a reduction in bone marrow B cells from an average of 20% in young mice to only 13% in 24-month-old mice (Figure 1A,B).

The total cell count of B cells was significantly reduced in Miz-1-deficient mice compared to control animals at all analyzed timepoints (Figure 1C). However, the total bone marrow cellularity is not altered between the groups (Figure 1C and Appendix A). Of note, expression of Mb1-Cre in Miz-1^wt^ mice did not alter bone marrow and B cell count and Miz-1^wt^ x Mb1-Cre^tg^ animals are indistinct from Miz-1^fl/fl^ x Mb1-Cre^wt^ (control) animals (Appendix A).

Investigation of different B maturation stages revealed a significant difference in early maturation stages (B220^+^IgM^−^) in young mice, whereas old mice showed only a slight, but not significant reduction in these cells (Figure 2A,B). Of note, B220^high^IgM^−^ cells were not significantly affected by aging nor by Miz-1 deficiency (Figure 2A,B). After the pre-pro B cell stage, all B cells express B220 and CD19 as markers for lineage commitment. Therefore, we included CD19 in our analysis of bone marrow-derived B cells. The frequency of immature (B220^+^CD19^+^CD93^+^CD43^−^IgM^+^) and recirculating (B220^+^CD19^+^CD93^−^CD43^−^IgM^+^) B cells was significantly reduced by more than 50% compared to control animals at all analyzed timepoints (Figure 2D,E). At 24 months, immature and recirculating B cells are barely detectable in Miz-1^ΔPOZ^ mice. Further, the absolute cell numbers of both populations were drastically decreased in Miz-1^ΔPOZ^ compared to control animals (Appendix A).

As previously described in aging mice, control animals showed a reduction in immature B cells by around 25% between 3 and 24 months, while recirculating B cells are only marginally affected in aged controls (Figure 2D,E) [52]. Further investigation showed that the frequency of pro-B cells was not different between Miz-1-deficient and control animals, but generally decreased with age, while cell numbers remained constant (Figure 3A,B and Appendix A). However, pre-B cells are significantly decreased in frequency by 5-fold from 9% in controls to only 1.8% in Miz-1^ΔPOZ^ mice at 3 months of age (Figure 3C). On average, both groups showed a decrease in pro- and pre-B cells of up to 50% between 3 and 24 months. The severe reduction in pre-B compared to pro-B cells in Miz-1^ΔPOZ^ mice increases the ratio between these two populations in favor of a pro-B cell accumulation (Figure 3A,D). Additionally, pre-B cell counts were significantly reduced in aged Miz-1^ΔPOZ^ mice compared to controls (Appendix A). However, *Zbtb17* and other critical transcription factors for B cell development and identity, such as *E2A*, *Ebf1* and *Pax5*, were not differentially expressed in aged B cell precursors of Miz-1^ΔPOZ^ and control mice (Appendix A). Only inhibitor of DNA binding 2 (*Id2*) was elevated in aged B cell precursors of Miz-1^ΔPOZ^ mice, but expression in immature and peripheral B cells was not altered in comparison to controls (Appendix A).

In summary, our data indicate that the deletion of Miz-1 in B lymphocytes of the bone marrow impairs the differentiation of late precursors from the pre-B cell stage on. Here, aged control mice exhibit a similar, yet a not so drastic, decrease during aging. Additionally, the amount of recirculating B cells is diminished and indicates further defects in peripheral lymphoid organs.

### 3.2. Miz-1 Is Critical for B Cell Homeostasis and Immunocompetence in the Periphery

To address consequences of the defective bone marrow B lymphopoiesis for peripheral B cells, we analyzed B cells from the spleen and Peyer’s patches (PPs). These organs, amongst others, host antigen-induced differentiation and proliferation of B lymphocytes and position them as a central player in immune response and regulation of lymphocyte production in adulthood [53,54].

Overall, Miz-1^ΔPOZ^ mice exhibit a significant reduction in splenic B cells (CD19^+^) in cell count and frequency by 50–70% compared to controls, independent of age (Figure 4A,B). Flow cytometry data showed a severe decrease in follicular (FO) B cells (CD19^+^CD93^−^CD21^med^CD23^+^) to less than 15% in young Miz-1^ΔPOZ^ compared to 35% of splenocytes in controls (Figure 4C,D). Both genotypes show further reduction in FO B cells by 10–15% with age, to an average of 2% in aged Miz-1^ΔPOZ^ mice and 20% in controls (Figure 4C). Interestingly, the frequency of marginal zone (MZ) B cells (CD19^+^CD93^−^CD21^+^CD23^−^) was significantly increased by 2-fold in young Miz-1^ΔPOZ^ mice and remains elevated in relation to controls at all analyzed timepoints (Figure 4E). This is consistent with total cell counts and parental frequency of FO and MZ B cells (Appendix A). As demonstrated in flow cytometry plots, we detected a reduction in FO B cells, while an atypical CD21^−^CD23^−^ population (hereafter: age-associated B cells or ABCs) increases in both genotypes with age (Figure 4C,F). The total cell number of ABCs is not different between the two genotypes (Appendix A). However, the B cell pool of aged Miz-1^ΔPOZ^ mice is significantly enriched for ABCs, whereas young animals are indistinguishable from one another (Figure 4G). Furthermore, splenic transitional B cell stages (T1–T3) are only partially affected. Transitional T1 (CD19^+^CD93^+^IgM^high^CD23^−^) B cells are significantly reduced at all analyzed timepoints in frequency and number in Miz-1^ΔPOZ^ mice (Appendix A). Transitional 2 (CD19^+^CD93^+^IgM^high^CD23^+^) B cells are reduced in aged Miz-1^ΔPOZ^ mice compared to controls, while T3 (CD19^+^CD93^+^IgM^low^CD23^+^) B cells seem to not be affected (Appendix A). Finally, splenic B1 (CD19^+^B220^low/med^CD43^+^CD5^+/−^) cells are not affected by deletion of Miz-1 using the Mb1-Cre recombinase (Appendix A).

A complex transcription factor regulatory network in B cells maintains their homeostasis and alterations of certain components have been described for aged mice. We analyzed the expression of key transcription factors in splenic B cells of control and Miz-1-deficient mice at young and aged timepoints. Independent of the age, we detected elevated expression of the Miz-1-binding partner *Myc* (Appendix A). Further, we found a decrease in *Ebf1* and *Pax5* expression by around 25–50% between 3 and 12 months in control and Miz-1^ΔPOZ^ mice, while transcript levels of E2A, the upstream regulator of *Ebf1* and *Pax5*, are not altered (Figure 4H,I and Appendix A). However, we could not detect a significant difference in gene expression between the two genotypes. Young Miz-1-deficient mice showed elevated *Id2* expression, while age-related upregulation of *Id2* and *Zbtb17* expression is indistinct compared to controls at 12 months of age (Appendix A). Nevertheless, we discovered elevated expression of the pro-inflammatory cytokine TNF-α by splenic B cells from aged Miz-1^ΔPOZ^ mice (Figure 4J). Consistent with this, analysis of splenic lymphocyte populations revealed a highly increased infiltration of CD8^+^ T cells and inflammatory monocytes into the spleen of aged Miz-1-deficient animals (Appendix A).

In conclusion, we detected a severe reduction in naïve B cells that are normally involved in immunological responses after activation to mediate a specific antigen response. Additionally, a hyper-inflammatory environment in Miz-1-deficient mice possibly hampers B cell homeostasis and response. This indicates that Miz-1^ΔPOZ^ mice may not mount proper immune responses after B cell activation.

### 3.3. Miz-1 Deficiency Favors Accumulation of Long-Lived B Cell Subsets after Antigen Challenge

Peyer’s patches, as immune sensors of the intestine, maintain the homeostatic balance between commensal microbial load and response to pathogenic microbiota in the gut [55,56]. The constant sensing of antigens from the intestinal lumen induces ongoing GC reactions and antibody-secreting cell differentiation for production of neutralizing IgA antibodies [57,58]. Investigation of these lymphoid organs in Miz-1^ΔPOZ^ mice showed a significantly reduced number of PPs on the intestinal mucosal membrane with a mean of 4.3 compared to controls with a mean of 6.3 by the age of 3 months, and this declined significantly during aging (Figure 5A). The linear regression further points towards a rapid decline for Miz-1^ΔPOZ^ mice. In comparison to controls, the cell count calculated in relation to the number of PPs in mice was significantly reduced by up to 90% in Miz-1^ΔPOZ^ and a reduction in CD19^+^ cells by more than 50% at all analyzed timepoints (Figure 5B and Appendix A). Correlated with this, control animals show a frequency of 55–65% of CD19^+^ cells in PPs, which is significantly reduced to only 20–35% in PPs of Miz-1-deficient animals, although no structural defects in PP architecture were detectable (Figure 5C and Appendix A). Consistent with the B cell reduction, Miz-1-deficient animals show a significant decrease in GC B cells by up to 3-fold at all analyzed timepoints in comparison to controls (Figure 5D,E). Consistent with this, we found a significant reduction in CD19^+^ and GC B cells after immunization with NP-CGG (Appendix A). Strikingly, GC B cells are almost absent in PPs from 24-month-old Miz-1^ΔPOZ^ mice (Figure 5D,E). However, while the amount of GC in relation to CD19^+^ cells remained constant in controls at all ages, Miz-1-deficient animals show a significant reduction in the GC B proportion (Figure 5F).

As GC B cells are precursors of the actual effector cells in the gut, IgA memory B cells, we asked whether Miz-1-deficient animals can produce immune-competent memory B cells (B_mem_) in the PPs. Interestingly, flow cytometry revealed a significant accumulation of terminally differentiated intestinal B_mem_ (CD19^+^CD38^+^IgA^+^) in Miz-1^ΔPOZ^ mice in comparison to control animals (Figure 6A,B). However, the amount of intraluminal IgA was significantly reduced in 3- and 12-month-old animals (Figure 6C). In response to environmental antigens, splenic B cells enter a GC reaction and differentiate into plasma or memory B cells. Consistent with the data from PPs, we found increased accumulation of splenic memory (CD19^+^IgD^−^IgG1^+^CD38^+^GL7^−^) B cells after immunization with NP-CGG and non-immunized aged Miz-1^ΔPOZ^ mice, while switched (CD19^+^GL7^+^CD95^+^IgD^−^IgG1^+^) B cells were largely absent in the spleen of Miz-1^ΔPOZ^ compared to control animals (Appendix A).

In conclusion, our data indicate that the immune response of Miz-1-deficient animals is compromised, due to decreased GC formation. Additionally, IgA^+^ B_mem_ cells are increased, and the decreased IgA antibody titer further supports this finding.

## 4. Discussion

Aging compromises the protective B cell response, as defective antibody responses increase and render aged individuals more prone for infection or other risks [59]. Effects on B cell physiology during aging, namely altered distribution of mature subsets and compromised activation, have been reported for aged mice and humans [42,59,60,61].

Overall, a reduction in bone marrow B cells is observed in aging mice [17,35,62]. In part, aging of the microenvironmental niche in the bone marrow contributes to reduced production rates, as cytokine production by aged mesenchymal progenitor cells is reduced compared to younger mice [63]. In agreement, we detected an age-dependent decrease in bone marrow B cells in control and Miz-1-deficient mice, while Miz-1-deficient B cells are highly susceptible to this reduction. We only use male mice to transmit the Mb1-Cre, since this has been used to ensure B cell specificity [64]. Therefore, the described effect is B cell intrinsic, because Mb1-Cre only affects the B cell lineage from the pro-B cell stage on [65].

As indicated, B cell subsets in the bone marrow and periphery are equally affected, and the severity of age-dependent effects is variable between individual mice. Murine early B (B220^+^IgM^−^) and pro-B cell numbers remain stable during aging, while maturation into pre-B and the immature B compartment is particularly affected by aging [17,52,66,67,68]. Miz-1-deficient animals display a more severe reduction in pre-B and immature B cells as compared to controls and this indicates that Miz-1 may interfere with B cell-intrinsic mechanisms to prevent premature loss of these populations. Both old and senescent B cells show an altered expression of the anti-apoptotic factor *Bcl2* and pro-B cells have an increased susceptibility to apoptosis [69,70]. Miz-1 was shown to regulate the p53-mediated DNA damage response and Miz-1-deficient pro-B cells undergo apoptosis due to a dysregulation of the anti-apoptotic factor *Bcl2* [46,71]. The role of Miz-1 during the p53-mediated DNA damage response in B cell development is not fully clear yet, but Miz-1 upregulates the transcription of *Rpl22*, which inhibits the translation of the p53 mRNA during VDJ recombination [72]. Transition from pro- to pre-B cell requires the accommodation of DNA damage during VDJ recombination. Therefore, insufficient expression of the anti-apoptotic factor *Bcl2*, a known target gene of Miz-1, and impaired inhibition of the p53-mediated DNA damage response lead to the reduction in pre-B cells [73]. In line with this, recent data showed that TP53INP1 deficiency, an inhibitor of p53, significantly increases frequency of pro-, pre- and immature B cells by maintaining STAT5 signaling and inducing *Pax5* expression for preservation of B lymphopoiesis in old mice [16].

*E2A* is a key regulator for the essential B cell transcription factor regulatory network of Ebf1, Pax5 and other downstream effectors maintaining B cell homeostasis and identity. The transcript levels of E2A are not altered in aged mice, which is consistent with our data. However, the magnitude of E47 reduction renders pro-B cells highly sensitive to a pre-B cell loss [28,36,37]. Mouse studies showed that like reduction of *E2A* expression, increased *Id2* expression, an inhibitor of all E2A protein variants, interferes with B cell development and renders pro-B cells highly sensitive to differentiation arrest. Here, Gfi-1 and Ebf1 repress *Id2* to maintain B cell differentiation and aged B cells were shown to have an increased expression of Id2, which is consistent with our data from B cell precursors [22,74,75]. Publicly available ChIP-Seq data also showed Miz-1 binding to *Id2* which is reduced after POZ-domain deletion [72]. Therefore, Miz-1 may participate in *Id2* repression, and it has been described as an interaction partner of Gfi-1 suppression of *Cdkn1a* and *Cdkn2b* [76,77]. Additionally, a reduced expression of *Ebf1* or *Pax5* is associated with decreased differentiation potential of old B cells, which can be restored with a constitutively active form of STAT5 or retroviral expression of *Ebf1* which rescues old B cells from differentiation arrest [78]. Interestingly, Miz-1 deficiency is associated with decreased *E2A*, *Ebf1* and *Pax5* expression in certain B cell precursor populations and may contribute to the premature loss of homeostasis [46].

Overall, accumulation of terminally differentiated and senescent B cells in aged mice and pro-inflammatory environments reduce output kinetics from the bone marrow and reflect a homeostatic pressure to protect against lymphopoietic exhaustion but limit potent responses towards new antigens [11,33,35]. Rejuvenation experiments showed that depletion of peripheral B cells in old mice reactivates BM B lymphopoiesis comparable to young animals, however, without restoring immune competence in vivo [33,79].

In contrast to the bone marrow, peripheral B cell counts remain stable in aged wildtype mice but show a severe shift in subset distribution characterized by exclusion of antigen-inexperienced cells [11,17,80]. Consistent with our data, a reduction in transitional T1–T3 B cell stages by more than 50% occurred between young and aged mice [52]. In old mice, FO B cells decrease and in parallel a new population of so-called age-associated B cells (ABCs) increases [80]. Of note, ABCs are likely progeny of FO B cells and phenotypically resemble an antigen-experienced, autoreactive or pro-inflammatory state [12,81,82]. ABCs are a mixed population with memory-like and anergic features, which preferentially develop in females and show accelerated accumulation during autoimmune diseases and persistent inflammation [29,61,82,83,84]. Corresponding to published data, we detected an increase in ABCs for both genotypes with age in female mice. We detected a comparable phenotype in males, yet not as severe as in females (data not shown). Opposite to FO B cells, their naïve precursor, ABCs, do not rely on BAFF-R-mediated survival signals [82]. The B cell pool of aged Miz-1^ΔPOZ^ mice is almost exclusively dominated by ABCs, while FO B cells are largely absent, indicating an amplified kinetic for ABC accumulation in these mice. In previous studies, expression of *Tnfrsf13c* (coding for BAFF-R/BLyS receptor 3) was shown to be decreased in Miz-1-deficient B cells and may render FO B cells prone to ABC development to escape selection pressure (unpublished). Furthermore, B cell clonotypes expressing low TACI levels, which also belongs to the BLyS receptor family, persist within the transitional B cell stage and enter the mature FO B cell pool in aged individuals [85].

In old mice, low-grade inflammation, increased secretion of TNF-α and loss of homeostasis hamper B cell responses. As mentioned before, expression of E2A inhibitor Id2 increases with age and correlates with reduced IGHV and D alterations in aged transitional and naïve B cells and diminishes antibody responses of old B cells [23,86]. We detected elevated expression of *Id2* is in young Miz-1^ΔPOZ^ which is described in aged B cells [22]. Although E2A transcripts are not affected, protein stability of E2A variants is additionally reduced by increased NOTCH signaling in aged mice [12,36]. We detected increased c-Myc expression in Miz-1^ΔPOZ^ mice independent of age, which can result from increased NOTCH signaling [87,88]. Both mechanisms result in reduction in *Ebf1* and *Pax5* during development, which is consistent with our data from aged mice of both genotypes. However, their expression is severely downregulated in aged Miz-1^ΔPOZ^. Initially, the increased *Id2* expression may be compensated to maintain homeostasis, however, the combination of initial Id2 upregulation and elevated NOTCH signaling possibly leads to early loss maintenance by *Ebf1* and *Pax5* and compromises homeostasis, further self-amplified by the increasing inflammatory environment with age in Miz-1^ΔPOZ^ mice.

Compromised Pax5 expression impairs CSR in aged and senescent B cells, resulting in poor antibody responses [21,24]. Further, decreased E2A mRNA stability mediated via the p38-MAPK signaling pathway and limited AID expression occur, which additionally contribute to impaired antibody responses in old mice [25,26,27,28]. Miz-1 was shown to act as a signal- and pathway-specific modulator or regulator (SMOR) inhibiting TNF-α-induced activation of c-Jun N-terminal kinase (JNK1) [89,90,91]. Miz-1 does not directly regulate the TNF-α-induced activation of p38, but as p38 is activated by varied stimuli, together with the pro-inflammatory milieu, Miz-1 may interfere with this MAPK-associated pathway to maintain CSR efficiency [91,92]. In line with these mechanisms and the overall reduction in GC B cells in Miz-1^ΔPOZ^, we detected reduced class-switched antibodies in these mice at all analyzed timepoints, while plasma IgM responses remained intact (data not shown). With age, the ability to mount antigen-specific responses decreases not only due to decreasing FO B cell numbers, but also decreased GC and compromised GC integrity [93]. Consistently, in Peyer’s patches—which are sites of chronic GC reactions—we detected reduced GC B cells and accelerated reduction in GC B cells in Miz-1^ΔPOZ^ mice with age in comparison to control animals. Of note, Miz-1 is important to maintain a GC reaction after induction of genotoxic stress by DNA lesions, as it represses *Cdkn1a* and cell cycle arrest in cooperation with Bcl6, while Bcl6 represses *TP53* to impede the p53 response [94,95]. Age-associated changes in T or follicular dendritic cells as part of the GC reaction may further amplify sub-optimal antibody responses [96].

We describe the accumulation of memory B cells after antigen encounters at the expense of GC B cells and class-switched antibodies in Miz-1-deficient mice. This is also evident in spleens of young Miz-1^ΔPOZ^ mice and indicates predisposition to altered differentiation. Recently published data described GC B cells lacking functional Miz-1/Myc complexes with an enhanced memory B cell expression profile as well as increased memory B cell differentiation [47]. Overall, we propose that loss of functional Miz-1 leads to an early accumulation of antigen-experienced and senescent cells in the periphery and a premature aging phenotype. The peripheral B cell pool of Miz-1^ΔPOZ^ mice is dominated by memory B and age-associated B cells at an early timepoint, which are potent producers of the pro-inflammatory cytokine TNF-α [12,31]. Consistently, we also detected increased TNF-α expression in Miz-1-deficient B cells, as described for old unstimulated B cells [32]. Increased TNF-α levels impair B cell responses in an autocrine manner, while inhibiting TNF-α improves AID and CSR in mouse B cells [32]. As Miz-1 plays a role in inhibiting TNF-α signaling, the redisposition of Miz-1-deficient B cells towards potent TNF-α-producing cells and TNF-α upregulation in B cells may amplify the B cell aging via an autocrine feedback mechanism in these mice.

Although we detected an increase in MZ B cells in Miz-1^ΔPOZ^ mice, control and Miz-1-deficient mice show a decrease in this population with age. Reports on age-associated effects on MZ B cells are contradictory, being either expanded or reduced in old mice and further show high variability [32,52,97]. This is possibly a result of different genetic backgrounds or other environmental stimuli which can lead to MZ B accumulation [32,52,98]. As we detected only a minor increase in MZ B cell number, we propose that MZ B cell development is not affected by Miz-1 deficiency, but the overall increase might be a result of a pro-inflammatory milieu induced by premature aging.

Aside from B cell-intrinsic mechanisms, the pro-inflammatory environment and accumulation of inflammatory lymphocytes may contribute to the loss of B cell homeostasis within the first year.

Here, we propose that transcription factor Miz-1 interferes at several stages of B cell-intrinsic aging mechanisms and most likely the B cell-specific loss of Miz-1 accelerates B cell aging in a collaborative fashion with B cell-intrinsic and -extrinsic mechanisms.

## 5. Conclusions

B cell-specific Miz-1 deficiency leads to loss of homeostasis and accumulation of antigen-experienced and senescent B cells by cell-intrinsic mechanisms, which compromises immune responses. Additionally, this predisposes towards a pro-inflammatory environment and amplifies the premature reduction in B cells and leads to loss of immunocompetence.

## Figures and Tables

**Figure 1 biology-11-00504-f001:**
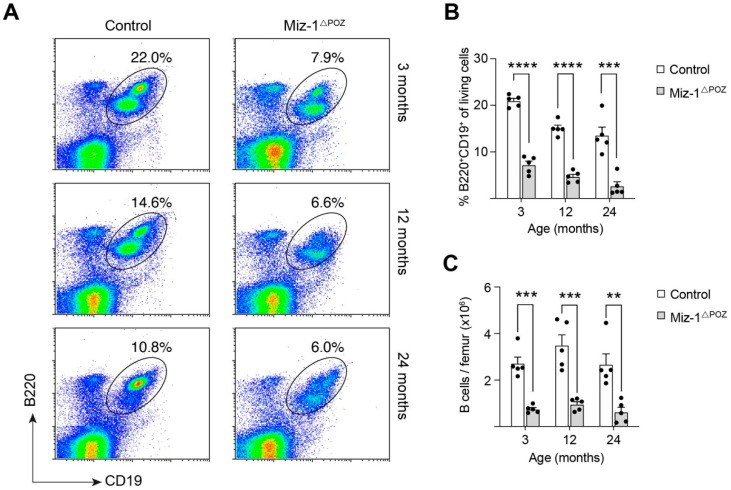
Age-related decline in bone marrow B cells. (**A**) Flow cytometric analysis of bone marrow cells from control and Miz-1^ΔPOZ^ mice. Cells were analyzed with antibodies for B220 and CD19. Cells are pre-gated for debris and doublet exclusion. Percentages in dot plots are given for the respective gates. (**B**) Frequency of B cells (B220^+^CD19^+^) from bone marrow of control and Miz-1^ΔPOZ^ mice. (**C**) Quantification B cells (B220^+^CD19^+^) from bone marrow of control and Miz-1^ΔPOZ^ mice, corrected to the number of living cells. One point represents one mouse. Data are expressed as mean + SD (*n* = 5). Student’s unpaired *t*-test: ** *p* < 0.01 *** *p* < 0.001, **** *p* < 0.0001. For (**A**), plots are representative of at least five independent experiments.

**Figure 2 biology-11-00504-f002:**
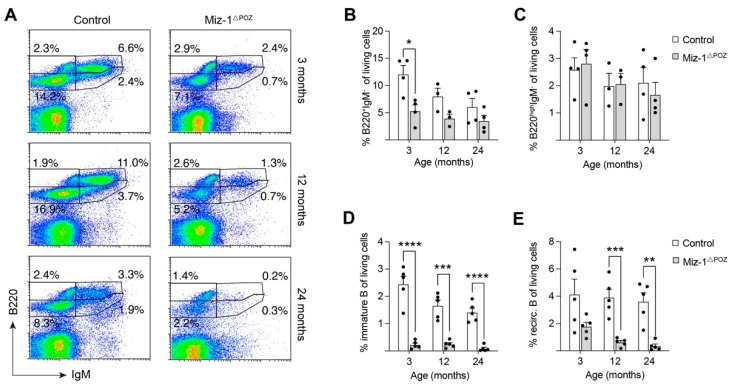
Decrease in immature and recirculating B cell subsets in the bone marrow with age. (**A**) Flow cytometric analysis of bone marrow cells from control and Miz-1^ΔPOZ^ mice. Cells were analyzed with antibodies for B220 and IgM. Cells are pre-gated for debris and doublet exclusion. Percentages in dot plots are given for the respective gates and were corrected to the number living cells. (**B**) Frequency of early B cells (B220^+^IgM^−^). (**C**) Frequency of B220^high^IgM^−^ B cells in the bone marrow. (**D**) Frequency of immature B cells (B220^+^CD19^+^CD93^+^CD43^−^IgM^+^) in the bone marrow of control and Miz-1^ΔPOZ^ mice at indicated ages. (**E**) Frequency of recirculating B cells (B220^+^CD19^+^CD93^−^CD43^−^IgM^+^) in the bone marrow of control and Miz-1^ΔPOZ^ mice at indicated ages. One point represents one mouse. Data are expressed as mean + SD (*n* ≥ 3). Student’s unpaired *t*-test: * *p* < 0.05, ** *p* < 0.01 *** *p* < 0.001, **** *p* < 0.0001. For (**A**), plots are representative of at least five independent experiments.

**Figure 3 biology-11-00504-f003:**
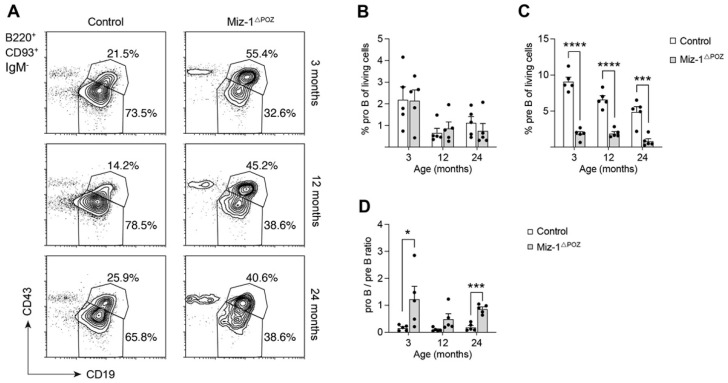
Miz-1 is important for the transition of pro- to pre-B cells during development. (**A**) Flow cytometric analysis of bone marrow cells from control and Miz-1^ΔPOZ^ mice at different ages. Percentages in plots are given for the respective gates and are quantified as percentage of B220^+^CD93^−^IgM^−^ cells. (**B**) Frequency of pro-B cells (B220^+^CD19^+^CD93^+^CD43^+^IgM^−^) from the bone marrow. (**C**) Frequency of pre-B cells (B220^+^CD19^+^CD93^+^CD43^−^IgM^−^) from the bone marrow. (**D**) Ratio of pro- to pre-B cells from control and Miz-1^ΔPOZ^ mice at indicated ages. One point represents one mouse. Data are expressed as mean + SD (*n* = 5). Student’s unpaired *t*-test: * *p* < 0.05 *** *p* < 0.001, **** *p* < 0.0001. For (**A**), plots are representative of at least five independent experiments.

**Figure 4 biology-11-00504-f004:**
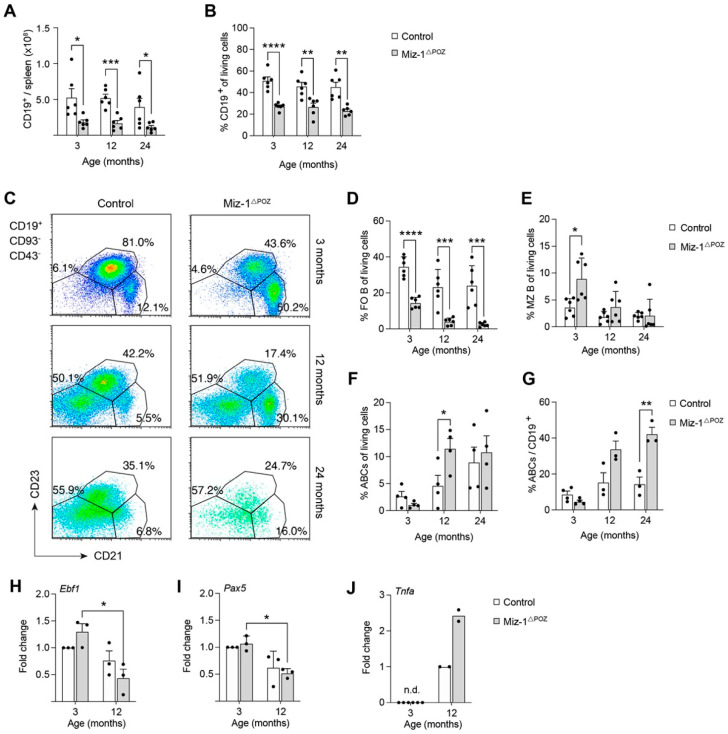
Miz-1 plays a role in fate decisions of peripheral B lymphocytes in aged mice. (**A**) Quantification of splenic CD19+ B cells. (**B**) Frequency of splenic CD19+ cells in control and Miz-1^ΔPOZ^ mice at different ages. (**C**) Flow cytometric analysis of splenocytes from control and Miz-1^ΔPOZ^ mice. Cells were analyzed with antibodies for CD19, IgM, CD93, CD21 and CD23. Percentages in plots are given for the respective gates as percentage of CD19^+^CD93^−^CD43^−^ cells. (**D**) Frequency of splenic FO B cells (CD19^+^CD93^−^CD21^med^CD23^+^). (**E**) Frequency of splenic marginal zone B cells (CD19^+^CD93^−^CD21^+^CD23^−^). (**F**) Frequency of age-associated B cells (CD19^+^CD93^−^CD21^−^CD23^−^). (**G**) Frequency of splenic ABCs (CD19^+^CD93^−^CD21^−^CD23^−^) relative to CD19^+^ cells. (**H**,**I**) Expression of *Ebf1* and *Pax5* in splenic B cells from young (3 months) and aged (12 months) control and Miz-1^ΔPOZ^ animals determined by qRT-PCR. Expression was normalized to young B cells of control animals. (**J**) Expression of *Tnfa* in splenic B cells from young (3 months) and aged (12 months) control and Miz-1^ΔPOZ^ animals determined by qRT-PCR. Expression was normalized to expression in aged control B cells. n.d., not detectable. One point represents one mouse. Data are expressed as mean + SD (*n* ≥ 2). Student’s unpaired *t*-test (**A**–**G**) and two-way ANOVA (**H**–**J**): * *p* < 0.05, ** *p* < 0.01, *** *p* < 0.001, **** *p* < 0.0001. For (**C**), plots are representative of at least five independent experiments.

**Figure 5 biology-11-00504-f005:**
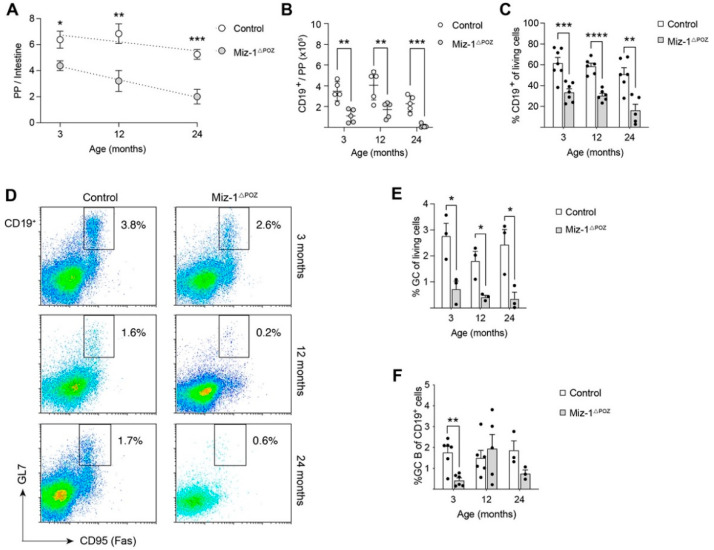
Non-functional Miz-1 leads to reduction in intestinal immunity. (**A**) Number of macroscopic Peyer’s patches (PPs) per intestine. Data are expressed as mean ± SEM (*n*
≤ 5). (**B**) Cell count of B cells (CD19^+^) from the intestine. Cells were counted and calculated to the number of PP/intestine. (**C**) Frequency of CD19^+^ cells in PPs analyzed by flow cytometry. (**D**) Flow cytometric analysis of lymphocytes from PPs of control and Miz-1^ΔPOZ^ mice. Cells were analyzed with antibodies for CD19, GL7 and CD95 (Fas). Percentages in plots are given for the respective gates as percentage of CD19^+^ cells. (**E**) Frequency of GC B cells (CD19^+^GL7^+^CD95+) in PPs from D. (**F**) Frequency of GC B cells in relation to CD19^+^ cell from PPs. Each point represents one mouse. Data are expressed as mean + SD (*n* ≥ 3). Student’s unpaired *t*-test: * *p* < 0.05, ** *p* < 0.01, *** *p* < 0.001, **** *p* < 0.0001. For (**D**), plots are representative of at least three independent experiments.

**Figure 6 biology-11-00504-f006:**
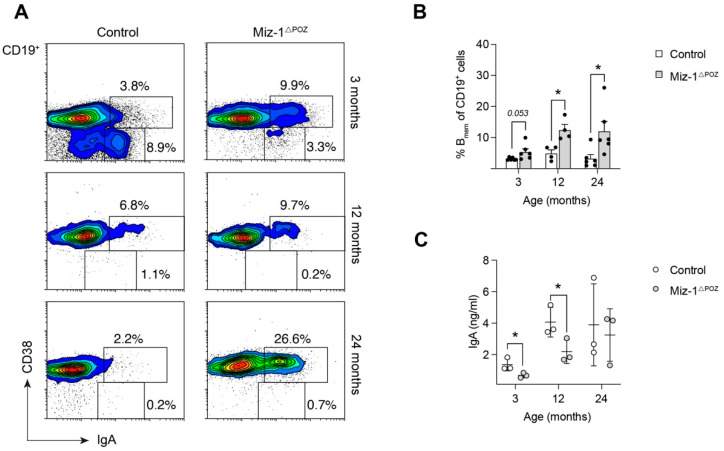
Antigen challenge boosts memory B cell differentiation at mucosal sites. (**A**) Flow cytometric analysis of single cells from macroscopic Peyer’s patches of control and Miz-1^ΔPOZ^ mice at different ages. Percentages in plots are given for the respective gates as percentage per parent (CD19^+^). (**B**) Frequency of memory B cells (B_mem_, CD19^+^CD38^+^IgA^+^) in relation to total CD19^+^ cells in PPs. (**C**) ELISA for fecal IgA from non-immunized control and Miz-1^ΔPOZ^ mice. Analysis was performed in technical duplicates and each point represents the mean for one mouse. Each point represents one mouse. Data are expressed as mean + SD (*n* ≥ 4). Student’s unpaired *t*-test: * *p* < 0.05. For (**A**), plots are representative of at least four independent experiments.

## Data Availability

Not applicable.

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
