# Peer review of "Myc-Interacting Zinc Finger Protein 1 (Miz-1) Is Essential to Maintain Homeostasis and Immunocompetence of the B Cell Lineage"

_biology, 2022, doi:10.3390/biology11040504_

Round 1
Reviewer 1 Report
Paper is well described and findings are interesting.
It will be interesting knowing if the authors testing also possible peripheral blood B cell lymphopenia?
Reviewer 2 Report
Authors investigated B-cell development in Miz-1 deficient mice. They report normal femoral bone marrow pro-B cell levels but a drastic decrease from the pre-B cell stage. The consequences of Miz-1 deficiency in periphery (spleen and Peyer’s patches) are reduced levels of follicular B-cells (in young and old mice) and elevated levels of marginal zone B-cells (in young mice) and age-associated B-cells (in old mice).
Results are clearly presented with convincing flow cytometric analysis and sufficient mice to validate statistical significance. Overall, I feel that the authors have delivered a technically very solid piece of work and applaud their efforts to make this work understandable for the greatest number. The paper is very descriptive but this remark is in no way detrimental to the reviewer.
Minor points:
Take care: The first paragraph under “3. Results” should be deleted.
Did the authors look to see if the Miz-1 deletion affected the T4/T8 and monocyte populations in bone marrow and spleen?
Authors always corrected their results to the number of living cells. Are percentages of dead cells identical in the 2 groups?
The discussion seems very long to me. A more focused discussion would be a plus for this paper.
Reviewer 3 Report
The manuscript from Piskor and Kosan describes the importance of Miz-1 transcription factor in the maintenance of immunocompetence of aging B lymphocytes. Miz-1 is a key factor to maintain B cell homeostasis through its partnering with Myc and its mediated regulation of IL7-R pathway and typical early B cell genes (Tcf3 or Ebf1), as reported in previous publications by the corresponding author of the manuscript (Kosan et al. Immunity 2010). The murine model used here has previously demonstrated the role of Miz-1 in early B cell development. Here, the authors focus on how Miz-1 deficiency alters B cell immunocompetence through aging, highlighting the premature B cell aging of Miz-1ΔPOZ mice (from now on, Miz-1 deficient mice)
Throughout the manuscript, the authors fully characterize B cell populations in bone marrow, spleen and Peyer patches (PP) at distinct time points of aging (3, 12 and 24 months). Data are presented in an elegant and solid manner and the manuscript is easy to follow. However, I believe that the manuscript would strongly benefit from the incorporation of further techniques (beyond flow cytometry) and a deeper insight into the molecular mechanism underlying Miz-1 function in aging B cells.
The specific points that need to be addressed are listed below.
Major points
- The authors use Miz-1fl/fl mice without Mb1-Cre recombinase expression as control for the experiments. It would be more appropriate to use Miz-1WT/fl or Miz-1WT/WT animals with Cre-recombinase as control, since it would eliminate the potential differences due to non-target effects of Cre recombinase. Have the authors checked that Cre recombinase has no off-target effects? Have they tested whether Miz-1WT/fl Mb1-Cre or Miz-1WT/WT Mb1-Cre have similar B cell counts than Miz-1fl/fl mice without Cre recombinase?
- In lines 148-149, the conclusion of B cell counts in bone marrow analysis should be rephrased. The authors state that total B cells in Miz-1 deficient mice are lower at all time points and decline even further with aging but, looking at data, the proportion is maintained constant. In the plots shown in Figure 1A, differences are even bigger at 3 months than at 24 months. The effect of Miz-1 in aging B cells is not that evident in bone marrow analysis.
- Which is the difference between data in Figures 2D and 3C? In both cases, looking at figure legends, it seems that the same staining has been used for bone marrow cells. Despite in Figure 2D are labelled as “% of immature B of living cells” and in Figure 3C as “% of pre-B of living cells”, it is difficult to understand whether they refer to different populations.
- The main differences in terms of B cell counts in aging mice (12-24 months) compared to young mice (3 months) are found in recirculating B cells (CD93 negative), as shown in Figure 2. Why Figure 3 only includes data on CD93+ cells? The pro-B to pre-B cell transition should be also analyzed in CD93- cells to corroborate the effect of Figure 2E in recirculating B cells.
- Flow cytometry plots shown in Figure 4C are not representative of the chart shown in Figure 4D. I understand that figure 4C shows the % calculated on the total of living cells. However, I think that a chart showing the % of FO B cells calculated on the total number of CD19+ CD93- CD43- cells should be included.
- The authors should provide data on CD19+ B220+ cell counts in the spleen, as performed in Figure 1A for bone marrow. Otherwise, figures like 4G are difficult to interpretate, since calculations are done only on the total of CD19+ cells. Is the total of CD19+ cells reduced in the spleen of Miz-1 deficient mice? Are there differences between different time points?
- Regarding the experiments on spleen and Peyer patches (PP), have the authors tried to induce GC reaction in the spleen? This can be easily performed by immunization with i.v. injection (normally intra-tail) of sheep red blood cells. The use of PP as a context for B cells undergoing GC reaction is interesting, but B cells in PP may present different behavior than in other secondary lymphoid organs (i.e. intestine microbiome, feeding conditions). For this reason, I believe that data from PP should be corroborated with splenic GC induction.
- The quantification of total number of Peyer patches in WT and Miz-1 deficient mice should be corroborated by staining in IHC from small intestine. They can also be visualized at macroscopic level. Images showing the differences between PP formation in both mice would be more informative than the chart in Figure 5A.
- In Figure 5F, the total number of CD19+ cells in PP should be provided, either by chart or by flow cytometry plots, as requested before for Figure 4G. Thus, we could visualize the potential differences in total B cell number in PP.
- Figure 6C seems to be contradictory with previous figures. The production of IgA by B cells at PP seems to be impaired under Miz-1 deficiency at 3 and 12 months of age. However, Miz-1 deficient mice acquire a similar capacity to secrete intraluminal IgA at 24 months of age. How do the authors explain this phenotype?
- Despite Miz-1 deficient mice have been described in previous publications, the authors should show how Miz-1 protein is affected by the deletion of POZ domain in B cells. Is this deletion affecting Miz-1 capacity to bind Myc or is it also affecting its expression? Protein or mRNA levels in B cells obtained from different organs (bone marrow, spleen or PP) should be provided. In case the expression is not altered, lack of binding to Myc should be shown (i.e. co-IP assay) to demonstrate the effect of POZ domain deletion in these populations.
- Several proteins and transcription factors involved in B cell development are mentioned throughout the manuscript (Id2, E2A, E47, Ebf1,…). How is the expression of these key factors in Miz-1 deficient mice? Again, quantitative real-time PCR or Western Blot assays would help to visualize the molecular mechanism underlying Miz-1 deficiency in B cells derived from bone marrow, spleen and PP.
- Would an exogenous forced expression of Gf1 or Ebf1 counteract the loss of Miz-1 function? Performing a knock-in murine model would be time consuming, but ex vivo cultures may be useful to overexpress these B cell factors and analyze whether immunosenescence of Miz-1 deficient B cells is reversed.
Minor points
- In the abstract, the abbreviation SLO should appear before (in line 20 would be appropriate).
- Line 60. The word “and” between “CSR” and “predispose” should be removed.
- Lines 137-139 should be removed since it seems to be part of the template file.
- Line 141. Instead of “effects”, it should be “affects”.
- Lines 358-359. It should be “…is likely to be…”, instead of “…is likely a to be…”.
Round 2
Reviewer 3 Report
The manuscript from Piskor et al. has been largely improved after the comments made in the revision. The authors have included valuable data in the main figures and the Supplementary Information, while they have properly answered and discussed some of the key points requested. Unfortunately, the time provided to perform experimental work requested was not enough.
I hope they can continue with this line of research in the future and provide answers to some of the unsolved questions about the role of Miz1 in B lymphocyte development.